# Aronia Melanocarpa: Identification and Exploitation of Its Phenolic Components

**DOI:** 10.3390/molecules27144375

**Published:** 2022-07-08

**Authors:** Theodora Kaloudi, Dimitrios Tsimogiannis, Vassiliki Oreopoulou

**Affiliations:** 1Laboratory of Food Chemistry and Technology, Department of Chemical Engineering, National Technical University of Athens, 5 Iroon Polytechniou, Zografou, 15780 Athens, Greece; theodorakaloudi@gmail.com (T.K.); ditsimog@chemeng.ntua.gr (D.T.); 2NFA (Natural Food Additives), Laboratory of Natural Extracts Development, 6 Dios st, Tavros, 17778 Athens, Greece

**Keywords:** black chokeberry, pomace extraction, anthocyanins, chlorogenic acid, spray drying encapsulation

## Abstract

The phenolic components of *Aronia melanocarpa* were quantitatively recovered by three successive extractions with methanol. They comprise anthocyanins (mainly cyanidin glycosides) phenolic acids (chlorogenic and neochlorogenic acids) and flavonols (quercetin glycosides). Approximately 30% of the total phenolic compounds are located in the peel and the rest in the flesh and seeds. Peels contain the major part of anthocyanins (73%), while the flesh contains the major part of phenolic acids (78%). Aronia juice, rich in polyphenols, was obtained by mashing and centrifugation, while the pomace residue was dried and subjected to acidified water extraction in a fixed bed column for the recovery of residual phenolics. A yield of 22.5 mg gallic acid equivalents/g dry pomace was obtained; however, drying caused anthocyanins losses. Thus, their recovery could be increased by applying extraction on the wet pomace. The extract was encapsulated in maltodextrin and gum arabic by spray drying, with a high (>88%) encapsulation yield and efficiency for both total phenols and anthocyanins. Overall, fresh aronia fruits are a good source for the production of polyphenol-rich juice, while the residual pomace can be exploited, through water extraction and spray drying encapsulation for the production of a powder containing anthocyanins that can be used as a food or cosmetics additive.

## 1. Introduction

*Aronia melanocarpa* (aronia), commonly known as black chokeberry, is a native plant of North America, but it is now cultivated worldwide. It belongs to the Rosaceae family and its berries have a deep purple color as they are rich in anthocyanins, as well as other phenolic compounds.

Aronia has higher total polyphenol content, the highest anthocyanin content and among the highest DPPH radical scavenging activity, compared to other berries [1,2]. Procyanidins have been identified as the major class of phenolic compounds [3]. More than 80% of the procyanidins consist of polymers with a polymerization degree higher than 10 [4], while (-)-epicatechin is the main monomer [3]. Anthocyanins are the phenolic compounds responsible for the dark purple color of the fruit. They amount to 25–50% of the total phenolic compounds [3,5]. Cyanidin 3-*O*-galactoside and cyanidin 3-*O*-arabinoside are the predominant anthocyanins, representing more than 90% of the total anthocyanins content, while cyanidin 3-*O*-glucosside and cyanidin 3-*O*-xyloside are present in low amounts [1,2,3]. Chlorogenic and neochlorogenic are the main phenolic acids of the fruit [3], while flavonols are the smallest class of phenolic compounds, amounting to less than 1.5% of the total phenolics, and comprising mainly quercetin derivatives [3,6,7].

Kulling [8] reviewed the components of the fruit and their correlation with potential health benefits. The biological activities of aronia polyphenols include antioxidant, antimutagenic, antidiabetic, cardioprotective, gastroprotective, hepatoprotective, anti-inflammatory, anti-carcinogenic, and geroprotective effects [9,10,11]. The high radical scavenging activity of *A. melanocarpa* was highly correlated with the total polyphenols content and moderately with the content of total anthocyanins [2,12]. Among anthocyanins, cyanidin 3-*O*-arabinoside possessed the highest DPPH radical scavenging activity, and also the highest activity against peroxidative and prooxidative enzymes, thus suppressing the production of reactive oxygen species (ROS), with a positive effect on cardiovascular function [13]. Additionally, cyanidin 3-*O*-arabinoside showed high activity against α-glucosidase [13], and may be responsible, together with procyanidins, for the prevention of diabetes-associated complications by the consumption of aronia juice [8].

The fresh fruit has a sour and bitter taste that is not pleasant to the consumer. Therefore, it is commercialized in a dry form but also as juice, nectar, wine, or jam [14,15,16]. Recent studies have examined the drying of aronia juice to produce a powder that can be added to functional foods, to upgrade their quality due to the beneficial health effects of its constituents, or to be used as a food colorant [17,18]. Additionally, extracts of the fruit, rich in anthocyanins and other phenolic compounds, can be used in food or pharmaceutical applications. Optimization of the extraction conditions has been reported in respect to extraction solvent, time, temperature, solid-to-liquid ratio and particle size [19,20,21]. Ultrasound irradiation generally facilitates the extraction and improves the yield of phenolic compounds [20,21]. Klisurova et al. [22] isolated anthocyanins from fresh aronia fruits and examined their co-pigmentation with other phenolic compounds or plant extracts, with the aim to enhance color and anthocyanin stability, so as to be used as additives in functional foods. The waste remaining from the preparation of juice products is rich in polyphenols, mainly anthocyanins [15], but the research about its utilization is very limited [20,23].

In Europe, a large part of the food waste results from the production of juices, since 3% of the total food waste concerns juicing by-products. However, research has mainly focused on the utilization of the juicing by-products of the major horticultural products, such as grapes, raspberries, citrus fruits, pomegranates and apples, while many other species including aronia need to be further investigated [23]. Due to consumer demand for functional foods, the cultivation of berries such as aronia is continuously augmenting. For example, Poland, which is the world’s largest producer, increased its production from 38,000 to 58,000 metric tons from 2004 to 2013 [11]. Thus, an increasing amount of fresh fruit or the by-products of industrial processing should be considered for exploitation. Moreover, aronia pomace is characterized by great morphologic diversity. It mostly consists of fruit skin, while smaller fractions of seeds, flesh and agglomerates of the above are also detected [24]. Despite extensive research on polyphenols in aronia fruit, extracts and juice, there is no scientific literature on the polyphenolic composition of fruit fractions. These data are considered very useful in the case of using aronia pomace as a potential source of polyphenol-rich formulations.

The present work focuses initially on the quantitative extraction and analysis of the phenolic compounds from the peels and flesh of aronia fruits, in order to identify them and quantify their distribution in the fruit. Additionally, a complete exploitation of the fruit was attempted through juice production and recovery of the bioactive components from the remaining pomace. A novel approach of fixed bed extraction with continuous solvent flow was applied, while water, at room temperature, was selected as the greenest solvent. Such a procedure could be easily upgraded to an industrial scale for the production of bioactive-rich extracts from aronia. The extracts were encapsulated in maltodextrin or its mixture with gum arabic, by spray drying, to obtain a powder that could be used as an antioxidant additive or colorant in food and cosmetics.

## 2. Results

### 2.1. Identification and Quantification of the Phenolic Components

#### 2.1.1. Recovery of Phenolic Compounds and Antiradical Activity

Aronia berries were peeled manually to obtain a fraction of peels, amounting to 14% of the fresh fruit, and a fraction of flesh and seeds (hereafter referred to as flesh), amounting to 86%. The percentage of moisture in the whole fruit was 76.7%, while for the peel and the flesh, the corresponding percentages were 67.6% and 77.7%, respectively. Both fractions were subjected to three successive extractions with methanol acidified by 0.5%, *v*/*v*, trifluoroacetic acid, in an ultrasonic bath. Methanol was selected as an extraction solvent because it shows high efficiency in the extraction of phenolic components and, thus, is commonly used for analytical purposes [3,4,15]. The acid was added to protect the anthocyanins, which are unstable phenolic compounds but are among the most valuable components of the berry. The solid-to-liquid ratio was kept constant at 1:20, as it was found to be the optimum value for total phenolic content (TPC) and anthocyanin extraction [19]. The antiradical activity of each extract was tested by the DPPH radical assay. The TPC and the antiradical activity of the extracts obtained by the three successive extractions of both fractions are presented in Figure 1A,Β.

The obtained TPC by the three extractions amounted to 150 ± 5 mg GAE/g dry peel, and 79 ± 3 mg GAE/g dry flesh, with the major part (76% and 74% for the peel and flesh, respectively) recovered by the first extraction, followed by 17% and 19% by the second, and 7% by the third extraction step. Similarly, the highest antiradical activity was obtained from the peels amounting to 111 ± 1 mg TE/g dry peel for the three extractions, compared to the flesh (55 ± 1 mg TE/g dry flesh for the three extractions). Several researchers have used two to four successive extractions for analytical purposes [25]. Our results indicated that three successive extractions were adequate for our experimental conditions.

#### 2.1.2. Phenolic Compound Analysis

All the extracts were analyzed by HPLC-DAD and representative chromatograms of the flesh extract, recorded at 320 nm (maximum absorbance of phenolic acids), 360 nm (flavonols absorbance maximum) and 520 nm (antocyanins absorbance maximum) are presented in Figure 2. Chlorogenic acid was identified with the use of the internal standard, while neochlorogenic acid was confirmed according to the UV spectrum, elution time in relation to chlorogenic, and literature data. More specifically, the UV spectrum of the peak with a retention time of 6.7 min is almost identical to the one of chlorogenic acid (R.t. = 11.0 min), while the retention time of the compound in the column is lower than that of chlorogenic acid, which agrees with numerous scientific publications, such as those by Cebulak et al. [5], Slimestad et al. [7], Sójka et al. [24] and Taheri et al. [26] for neochlorogenic acid.

The main anthocyanin, cyanidin 3-*O*-galactoside, was identified with the use of the internal standard. The chromatographic pattern of anthocyanins was compared with the others reported in the literature and it presented very similar characteristics with the respective chromatograms published by Jakobek et al. [1], Cebulak et al. [5], Bräunlich et al. [13], Vagiri and Jensen [15], Sójka et al. [24], Kapci et al. [27], Wangensteen et al. [28] and Szopa et al. [29]. In all cases, cyanidin 3-*O*-galactoside elutes first and in most cases shows the highest peak. Cyanidin 3-*O*-galactoside is systematically followed by one minor peak (cyanidin 3-*O*-glucoside), then the second major anthocyanin (cyanidin 3-*O*-arabinoside) and finally a second minor (cyanidin 3-*O*-xyloside). Exactly the same pattern was detected in the current research (Figure 2), while the UV spectra of the peaks were characteristic of anthocyanins. Therefore, the three peaks at 520 nm, following cyanidin 3-*O*-galactoside, were characterized as cyanidin 3-*O*-arabinoside, cyanidin 3-*O*-glucoside, and cyanidin 3-*O*-xyloside (Figure 2).

As far as the non-anthocyanin flavonoids are concerned, three minor peaks were characterized as flavonols, according to their UV spectra. Rutin (quercetin 3-*O*-rutinoside) was used as the internal standard, and matched the peak with a retention time of 29.1 min. The other two peaks, with retention times of 26.4 min and 28.3 min, presented similar UV spectra with rutin, and it could be assumed that the compounds are also 3-*O*-glycosides of quercetin. According to the previous research [5,12,30], most of the flavonols found in aronia are quercetin and isorhamnetin derivatives, while kaempferol is also detected. However, Lee et al. [31] identified three main flavonols, including the 3-*O*-glycosides of quercetin, namely quercetin 3-*O*-vicianoside, quercetin 3-*O*-glucoside and quercetin 3-*O*-rutinoside, with the specific elution order. The same profile of flavonol elution was detected in the current research, with the last of the three peaks having been identified as quercetin 3-*O*-rutinoside. Therefore, the other two flavonols with retention times of 26.4 min and 28.3 min were tentatively identified as quercetin 3-*O*-vicianoside, and quercetin 3-*O*-glucoside, respectively.

All the compounds that were identified in aronia (chlorogenic acids, cyanidin and quercetin derivatives) possess catecholic hydroxyls, i.e., the *o-*di-OH-benzene structure. It has been evidenced that the catecholic structure presents enhanced antiradical activity [32,33,34]. Therefore, it is easily understood why aronia berries present the highest antioxidant activity among many fruits [23]. Gruvonaite et al. [23] demonstrated that the potent scavengers of DPPH radicals are almost exclusively the compounds that possess catecholic hydroxyls, by applying the on-line HPLC-DPPH method to aronia extracts.

The analyses revealed that the peel and flesh extracts obtained by the three successive extractions contained the same phenolic components, although at different concentrations. The anthocyanins were almost quantitatively extracted by the first extraction (95% and 90% of the total content from the peels and flesh, respectively), while the second extraction recovered minor amounts (4% and 8% of the total content from the peels and flesh, respectively) and third less than 2%. The major compound among anthocyanins was the cyanidin 3-*O*-galactoside, followed by the cyanidin 3-*O*-arabinoside and in much lower concentrations, cyanidin 3-*O*-glucoside and cyanidin 3-*O*-xyloside were found in all extracts.

The highest amount of phenolic acids (85–86%) was obtained by the first extraction and the rest by the second, with traces found in the third extract. Chlorogenic acid was the most abundant, followed by neochlorogenic acid. Similarly, two successive extractions were sufficient for the recovery of flavonoids, with the first extracting 93% and 86% of the total content from the peels and flesh, respectively, and the second the rest.

#### 2.1.3. Phenolic Compound Partition in the Flesh and Peel of the Fruit

Table 1 presents the total amount of each component, obtained by the three successive extractions of either the peel or the flesh of the fruit, expressed on both fresh and dry bases of the respective fraction. The concentration in the fruit is calculated according to its peel and flesh content and is also presented in Table 1.

Considering the *w*/*w* percentages of peel and flesh in fresh aronia berries (14% peel and 86% flesh), the contribution of each fraction to the total phenolic content of the fruit was calculated. Anthocyanins were principally found in the peel of the fruit. More specifically, the peels contained 73% of the total anthocyanin content in the fruit. Regarding flavonols and phenolic acids, the peel contribution was lower, with the corresponding percentages being 46% and 22%, respectively. Anthocyanins are abundant in the peel of most fruits [35,36], but there is no published report about their distribution or that of the rest of the polyphenols in aronia peel and flesh.

Table 2 presents the concentrations of the main phenolic components of aronia reported in the literature. Rop et al. [37] detected the lowest polyphenol content (7.78–12.85 mg/g on a fresh basis), while the highest was recorded by Samoticha et al. (80.08 mg/g on a dry basis) [16]. This is still lower than the present study, which determined 94 mg/g on a dry basis. It should be mentioned that the content expressed on a dry basis is approximately four-fold higher than the content expressed on a fresh basis, depending on the moisture content of the fruit. Considering the data expressed on the same basis, differences are due to different cultivars but also to different extraction procedures. Cujic et al. [19] used dry aronia fruits and achieved an extraction yield of 27.8 mg GAE/g, dry basis, under optimum conditions, which is much lower than that obtained in our experiments. Although they used mild conditions (40 °C), drying may have caused some loss of phenolic compounds. This is further verified by the lower content of individual components obtained by all researchers when dried fruits were used instead of fresh ones (Table 2, values in brackets). Concerning the different cultivars, Wangensteen et al. [28] analyzed samples from three *Aronia melanocarpa* species and recorded up to two-fold discrepancies in polyphenol and anthocyanin concentrations. The extraction solvent and procedure are another cause of the observed differences. Ochmian et al. [38] used acidified methanol as a solvent and repeated three times their extractions in an ultrasonic bath. Their results in anthocyanins and polyphenols are very close to ours, where a relevant method was performed. Rop et al. [37] recorded lower concentrations of polyphenols, who also used methanol for their experiments but not ultrasound assistance. Ethanol, or aqueous-ethanol have also been reported as efficient solvents for aronia extraction in previous research. Wangensteen et al. [28] used boiling ethanol and a reflux condenser for their experiments and repeated the extractions twice, achieving similar recovery of the phenolic compounds.

Regarding phenolic acids and flavonols, there are also significant differences between the researchers. It should be noted, however, that flavonols are in all cases much lower than in our specimen, while the reported content of chlorogenic and neochlorogenic acids is similar, contrary to our results, indicating the double content of the former compared to the latter. The effect of solvent on the extraction of phenolic acids is also important. Ochmian et al. [41] achieved 1.21 mg/g recovery in their experiments using methanol as a solvent, while Dudonné et al. [42] used 100% ethanol and obtained about half of this value. The largest discrepancies occur in the case of anthocyanins. Indicatively, Rop et al. [37] detected 1.01–1.20 mg/g (fresh basis) of cyanidin 3-*O*-galactoside, which is the anthocyanin with the highest concentration in Aronia. On the contrary, Mayer-Miebach et al. [25] detected 4.10–4.40 mg/g of this particular anthocyanin in fresh berries, while the corresponding value in the present study was 4.0 ± 0.1 mg/g (fresh basis).

### 2.2. Juice Production and Analysis

Aronia juice was obtained by mashing and centrifugation and the yield varied between 49% and 53%, *w*/*w*, respectively, on a fresh fruit basis, which amounted to 0.43 L/kg and 0.47 L/kg, respectively. The solid content, reducing sugar content, TPC, antiradical activity, as well as the content of the phenolic compounds quantified by HPLC-DAD are presented in Table 3.

For comparative purposes, a sample of a commercial chokeberry juice product was also analyzed and the results are also presented in Table 3. It can be observed that the commercial product had similar total solids and reducing sugar concentrations to the juice prepared in the laboratory; however, the phenolic content was significantly lower. More specifically, it had a 59% lower polyphenol concentration, while the phenolic acids, flavonols and anthocyanins concentrations were 41%, 79%, and 96% lower than the laboratory prepared juice, respectively.

Four main antocyanins were detected in the juice, with cyanidin 3-*O*-galactoside showing the highest concentration, followed by cyanidin 3-*O*-arabinoside, in a similar manner to the fruit. In addition, chlorogenic acid showed a higher concentration than neochlorogenic. In addition to the three main flavonols detected in the fruit, four secondary peaks appeared in the juice samples, which were classified as flavonols, based on their UV spectra, but could not be identified. They were quantified as quercetin rutinoside equivalents and their total concentration is shown in Table 3.

Literature reports are also included in Table 3. Tolic et al. [47] analyzed 11 samples of natural commercial chokeberry juices and compared their content in polyphenols, anthocyanins and flavonols. They detected significant differences in the concentrations of polyphenols, non-flavonoids and flavonoids (3002–6639 mg GAE/L, 808–1527 mg GAE/L and 2180–4384 mg GAE/L, respectively); however, in the particular case of anthocyanins, the variances were much higher, as the concentrations ranged from 154 to 1228 mg CGluE/L. Anthocyanins are unstable compounds and can be affected by many factors, such as pH, light and oxygen. In addition, it has been found that several processes applied to the juice production, such as pressing, blanching and pasteurization, can play a very important role in anthocyanins’ preservation or degradation [27,48]. Our results showed agreement with these findings, as the juice prepared in the laboratory had an anthocyanin concentration of 1395 mg CGalE/L, while the commercial product contained only 57.2 mg CGalE/L. Finally, Jakobek et al. [6] also analyzed a natural aronia juice prepared in the laboratory and recorded significantly higher concentrations of polyphenols and anthocyanins (9154 mg GAE/L and 2637 mg CGluE/L, respectively) compared to our results. This may arise from the fact that juicing preparation included a juice extractor or may be associated with the cultivars of raw material [37]. Heating of the fruit mash (up to 60 °C) before juice production is a way to increase the TPC and anthocyanin content of the juice. However, Denev et al. [14] observed that a further increase to 80 °C had no significant effect on anthocyanin content but increased the TPC and proanthocyanidins in the juice, imparting an unpleasant, astringent taste.

Taking into account the TPC, anthocyanin, phenolic acid, and flavonoid content in the fresh fruit (presented in Section 2.1.3.), the average juice yield of 45.3 mL/100 g berries, and the contents presented in Table 3, it was calculated that 13% of the fruit TPC were transferred to the juice, 10% of the total anthocyanins, 66% of the phenolic acids, and 23% of the flavonoids. Therefore, a considerable amount, comprising mainly anthocyanins and flavonoids, remains in the pomace and could be further extracted for exploitation as food colorants, supplements etc.

### 2.3. Phenolic Compound Recovery through Pomace Extraction

#### 2.3.1. Extraction Kinetics

Wet fruit pomace is susceptible to microbial spoilage. Therefore, it needs drying if the extraction is not performed immediately after juice production. Thus, the pomace was dried, ground and kept at room temperature until further processing. The extraction was performed in a fixed bed column, with water acidified by citric acid (0.75%, *w*/*v*). Water was selected as the most environmentally friendly solvent and citric acid as a widely used, non-toxic acidifying agent that favors the extraction of anthocyanins [49]. The total extraction time was 180 min. The obtained TPCs in the extract at the exit of the fixed bed are plotted versus the extract volume in Figure 3A.

The concentration of TPC in the extract leaving the fixed bed rapidly increases during the first 10 min, as the solvent washes out the phenolic compounds located on the surface of the pomace particles [50]. Afterwards, the solvent has to penetrate into the particles, dissolve the phenolic compounds, and the latter will diffuse through the particles to the bulk volume of the solvent. Thus, the concentration decreases as the TPC of the pomace is decreasing, approaching exhaustion. The obtained curves in Figure 3A were fitted with the following simple mathematical equations presented below:TPC = 25,254 V + 1099.2  V < 0.050 L(1)
TPC = 111.15 V^−1^  V ≥ 0.050 L(2)
where TPC = total phenolic content at the exit of the bed in mg GAE/L and V = total volume of extract recovered in L.

The total area defined by the curves and the horizontal axis represents the mass recovery of polyphenols through the pomace extraction. By integrating each branch separately and ensuring that the resulting function will be continuous everywhere, the curve shown in Figure 3B was obtained. The equations that fit the curves of washing and diffusion stages in Figure 3B are as follows:TPC = 15,332.5 V^2^ + 1050.3 V  V < 0.050 L(3)
TPC = 9286.7 V^0.012^ − 8870  V ≥ 0.050 L(4)
where TPC = total phenolic content mass recovery in mg GAE and V = total volume of extract recovered in L. 

The mathematical model of the diffusion stage (Equation (4)) shows agreement with the power law model presented by Peppas et al. [51]. It is an empirical relationship that has been used successfully in many studies to describe the recovery of components from plant tissues. To evaluate the mathematical adjustment that followed, a sample of the final extract was directly analyzed. The total TPC recovery determined through the analysis (327 ± 11 mg GAE) was very close to the recovery predicted by the model (325.4 mg GAE).

The semi batch extraction, performed in a fixed bed reactor, has several advantages compared to a conventional solid/liquid extraction, performed in a continuous stirring vessel [52]. Pure solvent is continuously pumped through the solid bed; thus, the solute concentration in the solid matrix is always higher than that in the solvent, resulting in a higher mass transfer potential. Moreover, there is no need for the separation of the solvent from the exhausted solid at the end of the extraction, thus facilitating the procedure, reducing cost, and enabling easier scale up for industrial application. Modeling the extraction performance (Equations (3) and (4)) can be used for scale up purposes to predict the yield obtained as a function of the solvent volume.

#### 2.3.2. Extraction Yield

The pomace extract was analyzed by HPLC-DAD and the peaks separated were similar to the fruit analysis. The content of each class of compound is summarized in Table 4. Taking into account the fresh fruit concentration in phenolic components presented in Section 2.1.3., the pomace yield obtained by the experimental juicing (49–53% *w*/*w*) and the initial moisture content of the fruit and the pomace (76.7% *w*/*w* and 66.7% *w*/*w* respectively), the polyphenols recovered through pomace extraction can be converted to the dry fruit basis. The results are presented in Table 5, together with the results of the polyphenols recovered into juice.

It is evident that almost the total amount of the phenolic acids was recovered either in the juice or the pomace extract. On the contrary, only 26.3% of the fruit anthocyanins were recovered in both the juice and extract, and 69.5% of the flavonols. Therefore, a high amount of phenolic compounds, and especially anthocyanins and flavonoids, were either degraded during processing or still remained in the pomace. Blanching or heating of the mash are the processing steps in juice production that negatively affect the anthocyanin content in the pomace [15]. Such steps were not followed in our experimental procedure. On the contrary, degradation might have occurred during drying of the pomace before extraction. To test this hypothesis, the pomace extraction was repeated under the same conditions, using mashed fresh pomace instead of dried and the results are presented in Table 4, together with the dried pomace. As can be observed, the recovery of total phenolic compounds increased from 22.5 to 33.6, i.e., by 49%. There was no significant increase in the recovery of hydroxycinnamic acids and only a slight increase (14%) in flavonols. Nevertheless, the anthocyanins increased by 114%. Thus, it is confirmed that drying, even if mild, can cause severe degradation to anthocyanins, so the maintenance methods of anthocyanin-rich fruits must be further investigated.

The results described above indicate that an appreciable amount of the phenolic compounds of the fruit, which were obtained with successive methanol extractions, is not recovered by water and remains in the pomace. Water is less efficient as an extraction solvent compared to methanol or ethanol-water mixtures [23].

Galvan d’ Alessandro et al. [20] reported that ethanol in 50% water increased by three-fold the yield of anthocyanins and by two-fold the yield of TPC obtained from aronia pomace extraction. Nevertheless, water was used in our experiments as the most environmentally friendly and cheap solvent. Moreover, aqueous extracts can be subjected to spray drying encapsulation without any pre-treatment, while ethanol needs to be removed and re-cycled to ensure feasibility of the process. The yield could be increased by further prolonging the extraction. The final solvent-to-solid ratio in our experiments was 1:18 (g pomace: mL extract) and an increase in the solvent volume passing through the fixed bed would result in higher recovery, as can be observed in Figure 3B. However, this would need a high water volume and result in a low concentration of bioactive compounds in the extract, and consequently higher costs for water removal. Extraction with hot water might be an alternative approach to increase the yield, as an increase in temperature from 20 to 70 °C in aqueous extraction increased the total phenolics and anthocyanins by approximately two-fold, but possibly degraded part of the latter, as the extraction time was prolonged [20]. It should be noted that the percentage of anthocyanins in our extract (dry pomace extraction) was 43% out of the total polyphenols, while the corresponding value for Galvan d’Alessandro et al. [20] was 20%. This increase is probably a consequence of the 0.75% addition of citric acid to the solvent.

### 2.4. Encapsulation of the Phenolic Extract

Maltodextrin is the most commonly used wall material in spray drying encapsulation because it has a low cost, high water solubility and low viscosity even at high concentration, an ability to form films and to encapsulate extracts or juices rich in phenolic acids and anthocyanins [52,53]. Its combination with gum arabic was reported to improve the retention of certain compounds, such as flavonoids and their glycosides, compared to maltodextrin alone [52,54,55]. Therefore, both maltodextrin, and its combination with gum arabic were tested in our experiments.

Vidović et al. [56] investigated the influence of spray drying inlet temperature and solids concentration in the feed mixture on the characteristics of the final product and on the yield of the process. They concluded that the optimal conditions for the encapsulation of anthocyanins from aronia pomace were 140 °C inlet temperature and solid concentration in the feed equal to 40% *w*/*w*. On the other hand, Bendaska et al. [17] preferred 30% solid concentration in the feed for the encapsulation of anthocyanins from aronia juice and Mahdavi et al. [54] achieved an efficiency of 92% for the encapsulation of anthocyanins from barbery (*Berberis vulgaris*) extract by adjusting a mixture of maltodextrin-gum arabic as the carrier material and 20% solids concentration in the feed. Based on the above findings, an inlet temperature of 140 °C and 30% solids concentration in the feed was chosen for our experiments. The feed flow was set to 4 mL/min to ensure the stability of the inlet and outlet temperatures.

Table 6 presents the results of the encapsulation yield and encapsulation efficiency with either maltodextrin or a combination of maltodextrin and gum arabic. Encapsulation yield expresses the percentage of polyphenols (or anthocyanins in particular) transferred from the liquid extract to the dry powder by spray drying. Encapsulation efficiency expresses the percentage of the above transferred compounds enclosed purely inside the microcapsules, and not at the surface. Maximum (100%) yield efficiency was considered the hypothetical case in which the total amount of polyphenols contained in the extract would be quantitatively recovered by spray drying, enclosed exclusively inside the microcapsules of the dry powder.

The results indicate that the encapsulation of polyphenols by spray drying performed very well for both carriers. The two carriers in combination with the conditions set provided equally high encapsulation efficiencies in polyphenols and in particular anthocyanins. Although the combination of maltodextrin with gum arabic provided a higher yield in total phenolic components, and maltodextrin alone in anthocyanins (Table 6), the differences are not significant. The addition of gum arabic to maltodextrin has been reported to improve the retention of more hydrophobic components in the microcapsules, but not of the hydrophilic phenolic acids or anthocyanins [52,53]. This is due to its hydrophobic protein fraction that is covalently linked to the hydrophilic maltodextrin structure, providing an excellent emulsifying ability for the retention of hydrophobic substances, while not affecting the hydrophilic ones.

Additionally, the encapsulation efficiency was above 99%, indicating very good protection of the antioxidant compounds. These values are higher than those reported for anthocyanins [54,57]. Mahdavi et al. [54] reported higher encapsulation efficiency when a combination of maltodextrin with gum arabic was used instead of maltodextrin alone, attributed to the highly branched structure of gum arabic. It should be noted, however, that the anthocyanin content in their powder was much higher than in our experiments.

In conclusion, both carriers performed equally well. However, maltodextrin has some additional advantages over gum arabic (lower price, better repeatability, availability), so could be the preferred carrier to an up-scale application of the process.

## 3. Materials and Methods

### 3.1. Solvents and Reagents

Methanol and trifluoroacetic acid for the extraction of fractions were obtained from Fischer Scientific (Leicester, UK). For the pomace extraction, citric acid monohydrate was used as an acidifying agent, obtained from Penta Chemicals (Prague, Czech Republic). The carriers used in the microencapsulation process were maltodextrine 18–20 DE obtained from Astron Chemicals (Attica, Greece) and arabic gum powder from Nexira (Rouen, France). The materials used for the analysis of the samples were Folin Ciocalteu phenol reagent (2N) from Carlo Erba Reagents (Barcelona, Spain), sodium acetate 3-hydrate from Panreac Quimica SA (Barcelona, Spain), sodium carbonate anhydrous from Penta Chemicals (Prague, Czech Republic), sodium hydroxide from Panreac Quimica SA (Barcelona, Spain), 2,2-diphenyl-1-picryl hydrazyl (DPPH) from Sigma-Aldrich (Steinheim, Germany) and gallic acid (98% *w*/*w*) obtained from Acrōs Organics (Fair Lawn, NJ, USA). The standard compounds used were rutin trihydrate from Merck (Darmstadt, Germany), cyanidin galactoside from Extrasynthese S.A. (Genay Cedex, France) and chlorogenic acid from Fluka (Buchs, Switzerland). Water, acetonitrile and methanol for HPLC analyses of the samples were obtained from Fischer Scientific (Leicester, UK).

### 3.2. Plant Material

The fresh *Aronia Melanocarpa* berries used for the experiments were a commercial organic product (Pegasus, Thessaloniki, Greece), cultivated and harvested in Northern Greece in September 2021. The LOT number of the provided material was 211005, while a voucher sample has been kept at the Laboratory of Food Chemistry and Technology. The berries were subjected to manual peeling in order to separate the two fractions (peel and flesh-stones). Juice and pomace were prepared through mashing the berries using a moderate speed household blender and then centrifuging the pulp for 10 min at 8000 rpm. The wet residue was dried in a tumble dryer at 30 °C for 24 h, and ground with a moderate speed household blender.

### 3.3. Pretreatment Procedures

#### 3.3.1. Peeling and Grinding

The fresh aronia berries were peeled with laboratory forceps and divided into two fractions, bark and flesh-seeds. Prior to extraction, each fraction was ground using a porcelain mortar with glass pestle.

#### 3.3.2. Pomace Drying

Aronia pomace (juicing residue) was collected in plastic plates and dried at 30 °C for 24 h (Tauro Essiccatory, Vicenza, Italy). Afterwards, the dry material was ground at moderate speed for 60 s at room temperature, using a household blender (Tefal Optimo, Rumilly, France).

### 3.4. Extraction Procedure

#### 3.4.1. Solid/Liquid Ultrasound Assisted Extraction (UAE)

Crushed peels of aronia berries (4 g) were placed in a laboratory beaker and then 100 mL of solvent was added (solid/liquid ratio: 1/25 *w*/*v*). Methanol acidified with trifluoroacetic acid (0.5%) was used as the solvent. The beaker was immediately immersed in an ultrasonic bath (Elmasonic S, Elma, Schmidbauer, Germany), equipped with an ultrasonic frequency of 37 kHz. The temperature of the bath was maintained at 30 °C. After 40 min, the beaker was removed from the bath and allowed to rest for 5 min. The extract and solid residue were then separated by decantation and the solid residue was subjected to two additional successive extractions under the same conditions. Each extract was placed in a glass container and led separately for analysis. The same procedure was followed for the extraction of the flesh.

#### 3.4.2. Fixed Bed Semi-Batch Extraction

The pomace powder (25 g) was added to a 65 mL capacity, vertical, stainless steel fixed bed extractor covered by a small amount of filler (cotton). A peristaltic pump (Millipore, MA, USA) was connected to the inlet of the extractor in order to inject the solvent into the bed and a flow rate of 3 mL/min was set. The solvent entered at the bottom of the extractor, passed through the plant material and exited though the top. Deionized water acidified with citric acid (0.75%) was used as the solvent and a glass volumetric cylinder was placed at the outlet of the extractor to collect the extract (Figure 4). The extraction was performed at room temperature. The moment at which the first drop of the extract was received was set as t = 0. Samples (1 mL) were collected in Eppendorf tubes from the extractor outlet at t = 0, 5, 10, 15, 20, 30, 40, 60, 80, 100, 135 and 180 min. The extraction lasted 180 min and the final solid/extract ratio was 1/18 (including the 1 mL samples).

### 3.5. Juicing

Commercial scale juicing methods for fruit and berries, including aronia, include hot pressing (involving heat and enzymes) but also cold pressing (involving no heat or enzymes) [15]. In order to simulate a commercial scale juicing process in a laboratory, a batch of aronia fresh berries was mashed at moderate speed for 60 s at room temperature using a household blender (Tefal Optimo, Rumilly, France) and then centrifuged for 10 min at 8000 rpm (ThermoFisher Scientific, Osterode, Germany). The upper phase was further filtrated to obtain the juice, which was directly analyzed to determine its phenolic content. The residual pomace was dried for further processing.

### 3.6. Spray Drying Encapsulation

#### 3.6.1. Preparation of Feed Mixture

The parameters for the spray-drying encapsulation were selected according to literature reports [17,52] and laboratory experience. Regarding the feed mixture, the wall material was added to the pomace extract without any pre-treatment that could cause anthocyanin degradation. The dissolution was performed slowly, under continuous stirring and at a constant temperature of 30 °C, so that the final mixture was completely homogenized. The final feed mixture had 30% *w*/*w* solids concentration and a core: wall ratio of 6.2:100. Either maltodextrin (18–20 DE) or a mixture of maltodextrin-gum arabic 4:1, *w*/*w*, were used as wall material, while the solids of pomace extract assemble the core material.

#### 3.6.2. Spray Drying Encapsulation

Each mixture was fed to the spray dryer (Büchi B-191 Mini, Büchi Labortechnik AG, Flawil, Switzerland) under continuous stirring and room temperature. The air inlet temperature was set to 140 °C, feed flow rate was set to 4 mL/min and the atomization pressure was set to 5 bar. The outlet temperature of the powder is a dependent variable in this process and in our experiments was between 90 and 100 °C. The produced powder was collected and subjected to further analyses on the same day.

### 3.7. Analytical Procedures

#### 3.7.1. Determination of Total Phenol Content

The total phenolic content (TPC) of the samples was determined by the Folin–Ciocalteu reagent, using the method of Singleton et al. [58]. A UV-Vis spectrometer (T90+, PG Instruments, Leicestershire, England) was used to measure the absorbance at 765 nm. Duplicate measurements of each sample were performed and averaged. The results are expressed as gallic acid equivalents (GAE), through the construction of a calibration curve with the authentic reference compound (gallic acid).

#### 3.7.2. Antiradical Capacity

The antiradical capacity of the samples was determined by the DPPH radical assay, using A UV-Vis instrument (T90+, UV-Vis spectrometer, PG Instruments, Leicestershire, England). Samples (0.1 mL), appropriately diluted in methanol, were added to 3.9 mL of 6 × 10^−5^ M DPPH radical solution in methanol, and the absorbance at 515 nm was recorded after 30 min, according to the methodology reported by Brand-Williams et al. [59]. Duplicate measurements of each sample were performed and averaged. The results are expressed as Trolox equivalents (TE), through the construction of a calibration curve obtained with the authentic reference compound (Trolox).

#### 3.7.3. HPLC-DAD Analyses

The high-performance liquid chromatographic method (HPLC-DAD) described previously by Psarrou et al. [60] and Kanakidi et al. [52] was used. For the needs of the current research, the only modification was that, apart from 280 nm (universal wavelength) and 360 nm (detection of flavonols), two additional wavelengths were used, namely 320 nm and 520 nm for the accurate detection of phenolic acids and anthocyanins, respectively. Three standard compounds were used, i.e., quercetin 3-*O*-rutinoside, cyanidin 3-*O*-galactoside, and chlorogenic acid. The identification of the respective compounds in the extracts was performed with the use of the above as internal standards, while the quantification of the individual components and groups of relative compounds was based on their respective calibration curves. The identification of the rest compounds was tentative and performed according to UV spectra, retention times and comparison with the literature.

Each extract was diluted appropriately, in order for the peaks to be quantified to meet the ranges of the respective calibration curves of the respective authentic reference compounds.

#### 3.7.4. Encapsulation Yield and Efficiency

For the determination of the encapsulation yield and encapsulation efficiency, the method of Kanakidi et al. [52] was followed. The total content (TC) of the encapsulated compounds in the powder produced by the spray dryer was determined by dissolving 1 g of powder in 10 mL of deionized water and stirring with a vortex apparatus for 30 s. Ethanol (15 mL) was then added and stirred for 5 min to precipitate the wall material. The mixture was filtered through a micro filter (0.45 μm), and the obtained liquid was subjected to HPLC-DAD and Folin–Ciocalteu reagent analyses to determine the recovery of anthocyanins and total phenolic content, respectively. The yield (%) was determined as the ratio of the compounds in the microcapsule (dry basis) to their content in the feed solids (dry basis), multiplied by 100.

The content of encapsulated compounds on the surface of the powder (SC) was determined by adding 1 g of powder to 10 mL of a mixture of ethanol:methanol (1:1, *v*/*v*) and stirring with a vortex for 1 min. The sample was then filtered through a micro filter (0.45 μm), and the liquid was subjected to HPLC-DAD and Folin Ciocalteu reagent analyses. The content inside the microcapsule can be determined by subtraction, i.e., TC-SC, and, consequently, the encapsulation efficiency (%), by using the following equation: ((TC-SC)/TC) × 100%.

## 4. Conclusions

The fresh aronia fruits, examined in this study, were rich in anthocyanins, and also contained chlorogenic acids and a small amount of flavonols, in agreement with the literature reported results. Peels contain the major part of anthocyanins (73% of the total amount in the fruit), while flesh contains the major part of phenolic acids (78%). Two successive extractions with acidified methanol in an ultrasonic bath were adequate to recover more than 93% of the phenolic compounds from the fruit. Laboratory juicing resulted in a product amounting to 50% of the fresh fruit and containing 13% of the fruit TPC and 10% of anthocyanins. Thus, the pomace left as a by-product can be further extracted by water to obtain an extract rich in bioactive components. Fixed bed extraction is a simple procedure that is easy to up-scale for industrial application, and can be applied for the extraction of either dry or fresh pomace. Modeling of the extraction can be used to predict the yield obtained as a function of the solvent volume. The obtained extract can be effectively encapsulated in maltodextrin, by spray drying at mild conditions, to form a powder that can be added as a colorant or antioxidant supplement in food or cosmetics.

Aronia pomace, generated though juice production, is not currently used, although it is very rich in bioactive compounds. The results of this study can be applied for its exploitation to produce an extract or a powder that is rich in anthocyanins and other phenolic compounds and is suitable as a food or cosmetics additive.

## Figures and Tables

**Figure 1 molecules-27-04375-f001:**
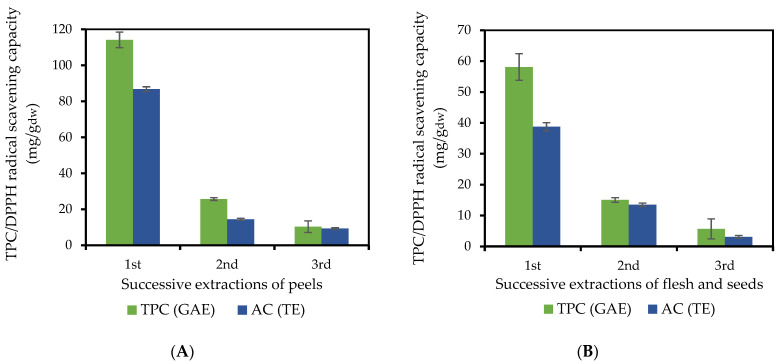
Total phenolic content (TPC, mg GAE/g_dw_) and DPPH radical scavenging capacity (AC, mg TE/g_dw_) obtained by successive ultrasound assisted extractions of (**A**) peels and (**B**) flesh and seeds of *A. melanocarpa* with acidified methanol. GAE: gallic acid equivalents, TE: Trolox equivalents.

**Figure 2 molecules-27-04375-f002:**
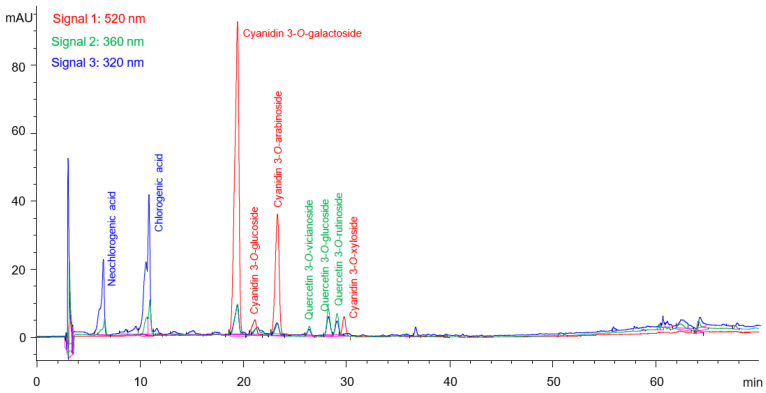
Chromatogram overlay of the first methanolic extract of aronia berries’ flesh at 520 nm, 360 nm and 320 nm.

**Figure 3 molecules-27-04375-f003:**
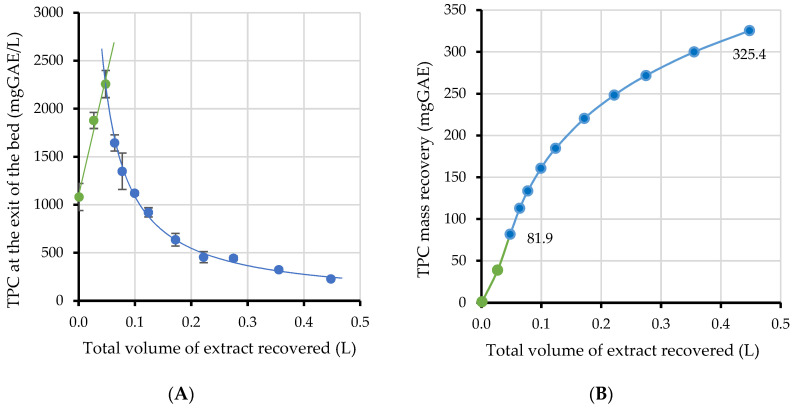
Recovery of phenolic compounds through pomace fixed bed extraction, at the exit of the fixed bed (**A**) and in total (**B**). The green line represents the wash out stage, and the blue line the diffusion stage of the extraction.

**Figure 4 molecules-27-04375-f004:**
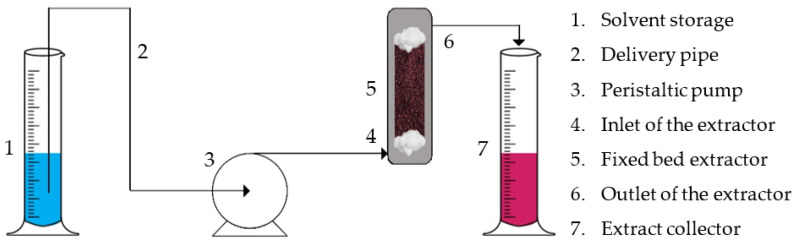
Schematic representation of the fixed bed semi-batch extraction arrangement.

**Table 1 molecules-27-04375-t001:** Concentrations of total phenolic compounds (mg GAE/g), antiradical capacity (mg TE/g), and individual phenolic groups determined by HPLC-DAD: anthocyanins (mg CGalE/g), phenolic acids (mg ChAE/g) and flavonols (mg QRE/g) in aronia peel, flesh and whole fruit.

	Peel (Fresh Vasis)	Peel (Dry Basis)	Flesh (Fresh Basis)	Flesh (Dry Basis)	Fruit (Fresh Basis)	Fruit (Dry Basis)
**Total phenolic content**	49 ± 2	150 ± 5	17.6 ± 0.3	79 ± 3	21.9 ± 0.4	94 ± 2
**Antiradical capacity**	35 ± 1	111 ± 1	12.3 ± 0.1	55 ± 1	15.6 ± 0.1	67.1 ± 0.4
**Anthocyanins**						
Cyanidin 3-*O*-galactoside	21.1 ± 1.0	65 ± 3	1.3 ± 0.1	5.7 ± 0.2	4.0 ± 0.1	17.4 ± 0.6
Cyanidin 3-*O*-arabinoside *	8.0 ± 0.3	25 ± 1	0.47 ± 0.02	2.1 ± 0.1	1.52 ± 0.05	6.5 ± 0.2
Cyanidin 3-*O*-glucoside *	1.8 ± 0.0	5.5 ± 0.1	0.08 ± 0.00	0.37 ± 0.01	0.31 ± 0.00	1.36 ± 0.02
Cyanidin 3-*O*-xyloside *	1.2 ± 0.1	3.8 ± 0.3	0.08 ± 0.00	0.38 ± 0.01	0.23 ± 0.02	1.05 ± 0.06
**Total**	32.1 ± 1.4	99 ± 4	1.9 ± 0.1	8.5 ± 0.3	6.1 ± 0.2	26.3 ± 0.7
**Hydroxycinnamic acids**						
Chlorogenic acid	0.80 ± 0.08	2.4 ± 0.2	0.45 ± 0.01	2.03 ± 0.06	0.50 ± 0.01	2.14 ± 0.04
Neochlorogenic acid *	0.40 ± 0.01	1.2 ± 0.0	0.24 ± 0.01	1.09 ± 0.03	0.27 ± 0.01	1.13 ± 0.02
**Total**	1.20 ± 0.07	3.7 ± 0.2	0.69 ± 0.02	3.12 ± 0.04	0.77 ± 0.13	3.27 ± 0.05
**Flavonols**						
quercetin 3-*O*-vicianoside *	0.22 ± 0.02	0.69 ± 0.05	0.068 ± 0.001	0.307 ± 0.001	0.09 ± 0.00	0.39 ± 0.01
quercetin 3-*O*-glucoside *	0.80 ± 0.00	2.48 ± 0.00	0.147 ± 0.002	0.659 ± 0.012	0.24 ± 0.00	1.02 ± 0.01
quercetin 3-*O*-rutinoside	0.76 ± 0.05	2.35 ± 0.14	0.130 ± 0.002	0.584 ± 0.008	0.22 ± 0.01	0.94 ± 0.03
**Total**	1.79 ± 0.03	5.5 ± 0.1	0.35 ± 0.00	1.55 ± 0.00	0.55 ± 0.01	2.32 ± 0.00

GAE: gallic acid equivalents, TE: trolox equivalents, CGalE: cyanidin 3-*O*-galactoside equivalents, ChAE: chlorogenic acid equivalents, QRE: quercetin rutinoside equivalents * tentative identification.

**Table 2 molecules-27-04375-t002:** Concentrations of total polyphenols (mg GAE/g), antiradical capacity (mg TE/g), anthocyanins (mg CGalE/g), phenolic acids (mg ChAE/g) and flavonols (mg QRE/g) in *Aronia melanocarpa* berries reported in the literature.

	Concentration (mg/g)	Reference
**Total phenolic content**	13.3, 10.79–19.21, 18.45–23.40, 7.78–12.85 (FW)	[27,28,37,38]
	78.49, 80.08, {19.6–27.8} (DW)	[3,16,19]
**Antiradical capacity**	11.00 (FW)	[27]
**Total anthocyanins**	2.52–4.47, 6.19 * (FW)	[28,39]
	39.17, {0.26–0.52} * (DW)	[16,29]
Cyanidin 3-*O*-galactoside	2.92, 4.10–4.40, 1.01–1.20, 4.24 * (FW)	[25,27,37,39]
	12.82, 2.21–14.50, {0.19–0.34} *, {0.40–0.85} (DW)	[3,19,26,29]
Cyanidin 3-*O*-araboniside	1.35, 1.90–2.10, 1.54 * (FW)	[25,27,39]
	5.81, 1.05–2.16, {0.07–0.12} *, {0.14–0.32} ** (DW)	[3,19,26,29]
Cyanidin 3-*O*-glucoside	0.078–0.27 *, 0.20 *, 0.19 * (FW)	[38,39,40]
	0.42, 0.049–0.191, {0.03–0.14} *, {0.07–0.14} * (DW)	[3,19,26,29]
Cyanidin 3-*O*-xyloside	0.13–0.14, 0.20 * (FW)	[25,39]
	0.52, ND, ND–0.150 (DW)	[3,16,26]
**Total hydroxycinnamic acids**	1.21, 0.63, 1.16 (FW)	[39,41,42]
	{2.87–3.93} (DW)	[29]
Chlorogenic acid	0.72–0.96, 0.69–0.74, 0.70 (FW)	[25,38,39]
	3.02, 3.32–6.42, {1.17–1.51} (DW)	[3,26,29]
Neochlorogenic acid	0.59–0.79, 0.56, 0.46 (FW)	[38,39,41]
	2.9, 2.16–6.54, {1.10–1.75} (DW)	[3,26,29]
**Total Flavonols**	0.71, 0.19–0.41, 0.34 (FW)	[7,39,43]
	{0.13–0.25} (DW)	[29]
Quercetin 3-*O*-galactoside	0.09–0.14, 0.28 ***, 0.11 (FW)	[39,44,45]
	0.36, {0.17–0.27} *** (DW)	[3,19]
Quercetin 3-*O*-glucoside	0.07–0.08, 0.043 ***, 0.07 (FW)	[39,44,46]
	0.21, {0.10–0.15} **** (DW)	[3,19]
Quercetin 3-*O*-rutinoside	0.05–0.06, 0.042 ***, 0.04 (FW)	[38,39,46]
	0.15, {0.31–0.42} (DW)	[3,19]
Quercetin derivatives unindentified	0.27 (DW)	[3]

GAE: gallic acid equivalents, TE: trolox equivalents, CGalE: cyanidin 3-*O*-galactoside equivalents, ChAE: chlorogenic acid equivalents, QRE: quercetin rutinoside equivalents, FW: expressed on fresh weight, DW: expressed on dry weight, * cyanidin glucoside equivalents, ** cyanidin arabinoside equivalents, *** quercetin galactoside equivalents, **** quercetin glucoside equivalents, { } values included in brackets regard analyses performed on dried berries.

**Table 3 molecules-27-04375-t003:** Concentrations of solid content (g/L), reducing sugars (g/L), total phenolic content (mg GAE/L), antiradical capacity (mg TE/L), anthocyanins (mg CGalE/L), phenolic acids (mg ChAE/L) and flavonols (mg QRE/L) in aronia juice.

	Laboratory Product	Commercial Product	Previous Research	References
**Total solid content**	197 ± 1	178.4 ± 0.1	13.42–21.54 **	[47]
**Reducing sugars**	53.0 ± 0.5	56.8		
**Total phenolic content**	6231 ± 159	2550 ± 83	9154, 3002–6639	[6,47]
**Antiradical capacity**	4265 ± 89	2257	3026–10,059	[47]
**Anthocyanins**				
Cyanidin 3-*O*-galactoside	987 ± 12	47.5	1816 *, 787	[3,6]
Cyanidin 3-*O*-arabinoside	308 ± 8	8.6	647 *, 324	[3,6]
Cyanidin 3-*O*-glucoside	54 ± 5	2.1	74.3 *, 28.1	[3,6]
Cyanidin 3-*O*-xyloside	45 ± 2	ND	99.8 *, 33.6	[3,6]
**Total**	1395 ± 17	58.2	2637 *, 154–1228 *	[6,47]
**Hydroxycinnamic acids**				
Chlorogenic acid	628 ± 38	127.1	415.9	[3]
Neochlorogenic acid	438 ± 5	161.0	290.81	[3]
**Total**	1116 ± 45	654		
**Flavonols**				
quercetin 3-*O*-vicianoside	38.8 ± 1.1		27.5	[3]
quercetin 3-*O*-glucoside	77.7 ± 0.2		31.2	[3]
quercetin 3-*O*-rutinoside	89.7 ± 3.1		49.7	[3]
Flavonols unindentified	62.8 ± 4		46.9	[3]
**Total**	269 ± 12	27.9 ± 4		

* Cyanidin glucoside equivalents, ** % *w*/*w*, ND = not detected.

**Table 4 molecules-27-04375-t004:** Recovery of total phenolic compounds (mg GAE/g), anthocyanins (mg CGalE/g), phenolic acids (mg ChAE/g) and flavonols (mg QRE/g) from aronia pomace. All expressed on a dry basis.

	Dried Aronia Pomace	Fresh Aronia Pomace
Total phenolic content	22.5 ± 0.06	33.6 ± 0.9
Anthocyanins	5.7 ± 0.2	12.21
Hydroxycinnamic acids	1.63 ± 0.03	1.69
Flavonols	1.51 ± 0.09	1.72

**Table 5 molecules-27-04375-t005:** Recovery of total phenolic content (mg GAE/g), antiradical capacity (mg TE/g), anthocyanins (mg CGalE/g), phenolic acids (mg ChAE/g) and flavonols (mg QRE/g) from aronia berries through juicing and dried pomace extraction. All expressed on dry fruit basis.

	Content in Berries	Recovery in Juice		Recovery in Dry Pomace Extraction	
	(mg/g Dry Fruit)	(mg/g Dry Fruit)	%	(mg/g Dry Fruit)	%
Total phenolic content	94 ± 2	12.3 ± 0.3	13.1	16.41 ± 0.04	17.5
Antiradical activity	67.1 ± 0.4	8.29 ± 0.17	12.4	11.12 ± 0.17	16.6
Anthocyanins	26.3 ± 0.7	2.73 ± 0.03	10.4	4.15 ± 0.15	15.9
Hydroxycinnamic acids	3.27 ± 0.05	2.15 ± 0.09	66.1	1.18 ± 0.06	35.1
Flavonols	2.32 ± 0.00	0.52 ± 0.02	22.5	1.10 ± 0.06	47.0

**Table 6 molecules-27-04375-t006:** Encapsulation yield and encapsulation efficiency with either maltodextrin or a 4:1 combination of maltodextrin and gum arabic.

	Yield (%)	Efficiency (%)
**T** **otal phenolic content**		
Maltodextrin	92.2 ± 2.6	99
Maltodextrin-gum arabic	96.6 ± 2.4	100
**Anthocyanins**		
Maltodextrin	91.4 ± 1.6	99
Maltodextrin-gum arabic	88.5 ± 2.0	99

## Data Availability

Not applicable.

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
