# Peer review of "Aronia Melanocarpa: Identification and Exploitation of Its Phenolic Components"

_molecules, 2022, doi:10.3390/molecules27144375_

Round 1
Reviewer 1 Report
The comments are as follows: 1. In the abstract section, the overall conclusion is missing. 2. The authors are encouraged to express more the importance of the study in the introduction section. Some statistics about aronia production/processing/consumption could be included. 3. Please, provide the moisture content and particle size of aronia fractions. 4. The extraction using methanol is not food grade, is it really necessary in this study? Why not use only the food grade solvents? If the purpose is to extract metabolites/compounds for food, cosmetic or pharma? 5. More details about fixed bed semi-batch extractor are needed. 6. The authors are encouraged to explain the relevance and future applications of the study. 7. Please, revise the text for some misprints.Author Response
Reviewer 1
The comments are as follows:
- In the abstract section, the overall conclusion is missing.
An overall conclusion has been added according to reviewer’s suggestion
- The authors are encouraged to express more the importance of the study in the introduction section. Some statistics about aronia production/processing/consumption could be included.
We would like to thank the reviewer for this suggestion. Some information about the importance of the study were added in the introduction.
- Please, provide the moisture content and particle size of aronia fractions.
Moisture content was added in the revised text. The fractions were ground manually, as described in the experimental section and particle size was not determined.
- The extraction using methanol is not food grade, is it really necessary in this study? Why not use only the food grade solvents? If the purpose is to extract metabolites/compounds for food, cosmetic or pharma?
Methanol was used for analytical purposes as explained in the text: “Methanol was selected as an extraction solvent because it shows high efficiency in the extraction of phenolic components and, thus, is commonly used for analytical purposes”. In the following sections, water was used to extract the phenolic compounds that could be used as food or cosmetic additives.
- More details about fixed bed semi-batch extractor are needed.
Details were added, together with a figure showing the experimental apparatus
- The authors are encouraged to explain the relevance and future applications of the study.
Comments about the relevance and future application of the study have been added to the conclusions section.
- Please, revise the text for some misprints.
The text has been carefully revised
Reviewer 2 Report
I recommend the reviewed manuscript Manuscript entitled: „Aronia melanocarpa: Identification and exploitation of its 2 phenolic components“ to be published in Molecules.
Experimental part is well set, experimental design consist of several parts, extraction with methanol, for extraction kinetics, extraction with water and encapsulation of the extract.
Methods are well described and results are well-presented and discuses.
Check of misspelling should be done.
Line 72: This sentence: „Following, the fresh fruits were mashed and 72 centrifuged to obtain juice, and the remaining pomace was used for phenolic compound extraction.” is not necessary in here. It is for methods section.
Line 183: Sentence is not clear enough: „Anthocyanins were principally found in the peel of the fruit, as the contribution of this fraction to the total fruit content was 73%.”
Line 378: Vidocic and al. [56]- Vidović is the correct
Author Response
Reviewer 2
I recommend the reviewed manuscript Manuscript entitled: „Aronia melanocarpa: Identification and exploitation of its 2 phenolic components“ to be published in Molecules.
Experimental part is well set, experimental design consist of several parts, extraction with methanol, for extraction kinetics, extraction with water and encapsulation of the extract.
Methods are well described and results are well-presented and discuses.
Check of misspelling should be done.
We would like to thank the reviewer for his comments. The manuscript has been carefully checked according to his/hers suggestion.
Line 72: This sentence: „Following, the fresh fruits were mashed and 72 centrifuged to obtain juice, and the remaining pomace was used for phenolic compound extraction.” is not necessary in here. It is for methods section.
The sentence was removed and replaced by “Additionally, a complete exploitation of the fruit was attempted through juice production and recovery of the bioactive components from the remaining pomace.”
Line 183: Sentence is not clear enough: „Anthocyanins were principally found in the peel of the fruit, as the contribution of this fraction to the total fruit content was 73%.”
The sentence was changed to: Anthocyanins were principally found in the peel of the fruit. More specifically the peels contain 73% of the total anthocyanins content in the fruit.
Line 378: Vidocic and al. [56]- Vidović is the correct
Corrected
Reviewer 3 Report
The manuscript entitled “Aronia melanocarpa: Identification and exploitation of its phenolic components” have described phenolic components of Aronia melanocarpa in methanol extract. Moreover, the extract was encapsulated in maltodextrin and gum Arabic by spray drying, with a >88% encapsulation yield and efficiency for both total phenols and anthocyanins. The aim of this study is important and topical, especially in the context of new antioxidants. Authors should correct manuscript according to the suggestion.
Minor Issues:
Figure 1: please explain 1,2,3 meaning on X axis
References: should be carefully checked and corrected according authors guides e.g. names of plants should be italicized, some names of journals Authors gives as abbreviations (for example Ref. no 15, 18, 22, 29....45-47)
Table 2: How explain such a differences between content of polyphenols, athocyanins and phenolic acids in Aronia Melanocarpa berries? In Table 2 I suggest include information how samples were prepared (e.g. ethanol or methanol were used for extracts preparation and which concentration?)
Author Response
Reviewer 3
The manuscript entitled “Aronia melanocarpa: Identification and exploitation of its phenolic components” have described phenolic components of Aronia melanocarpa in methanol extract. Moreover, the extract was encapsulated in maltodextrin and gum Arabic by spray drying, with a >88% encapsulation yield and efficiency for both total phenols and anthocyanins. The aim of this study is important and topical, especially in the context of new antioxidants. Authors should correct manuscript according to the suggestion.
Minor Issues:
Figure 1: please explain 1,2,3 meaning on X axis
1, 2, 3 are the successive extractions. For better understanding the numbers were replaced by 1st, 2nd, 3rd.
References: should be carefully checked and corrected according authors guides e.g. names of plants should be italicized, some names of journals Authors gives as abbreviations (for example Ref. no 15, 18, 22, 29....45-47)
We would like to thank the reviewer for this remark. All references were corrected in the revised text.
Table 2: How explain such a differences between content of polyphenols, athocyanins and phenolic acids in Aronia Melanocarpa berries? In Table 2 I suggest include information how samples were prepared (e.g. ethanol or methanol were used for extracts preparation and which concentration?)
The high differences are among the dried (values included in brackets) and fresh samples, as already commented in the manuscript. Some additional comments about the observed differences were included in the revised text.